# MTRec: Learning to Align with User Preferences via Mental Reward Models

**Mengchen Zhao**[1]    **Yifan Gao**[2]    **Yaqing Hou**[2*]   **Xiangyang Li**[3]    **Pengjie Gu**[4]
**Zhenhua Dong**[3]    **Ruiming Tang**[3]    **Yi Cai**[1]

[1]School of Software Engineering, South China University of Technology
[2]School of Computer Science and Technology, Dalian University of Technology
[3]Huawei Noah's Ark Lab    [4]Nanyang Technological University
{zzmc, ycai}@scut.edu.cn   {otz, houyq}@mail.dlut.edu.cn
{lixiangyang34, dongzhenhua, tangruiming}@huawei.com

## Abstract

Recommendation models are predominantly trained using implicit user feedback, since explicit feedback is often costly to obtain. However, implicit feedback, such as clicks, does not always reflect users' real preferences. For example, a user might click on a news article because of its attractive headline, but end up feeling uncomfortable after reading the content. In the absence of explicit feedback, such erroneous implicit signals may severely mislead recommender systems. In this paper, we propose MTRec, a novel sequential recommendation framework designed to align with real user preferences by uncovering their internal satisfaction on recommended items. Specifically, we introduce a mental reward model to quantify user satisfaction and propose a distributional inverse reinforcement learning approach to learn it. The learned mental reward model is then used to guide recommendation models to better align with users' real preferences. Our experiments show that MTRec brings significant improvements to a variety of recommendation models. We also deploy MTRec on an industrial short video platform and observe a 7% increase in average user viewing time.

## 1   Introduction

In interactive recommender systems, explicit feedback (e.g., ratings) is inherently sparse. Consequently, recommendation models predominantly rely on implicit signals (e.g., clicks) for training. However, such signals frequently fail to capture users' real preferences. For instance, clicking on a video may not indicate satisfaction with its content, while skipping a video could stem from prior exposure to similar content on other platforms rather than genuine dislike. These observations highlight a fundamental misalignment between recommendation models and users' real preferences.

To mitigate such misalignment caused by erroneous feedback signals, a natural approach is to incentivize users to provide explicit feedback. However, in real-world scenarios, users exhibit low propensity to offer such feedback due to cognitive burdens and interface constraints. Prior studies treat erroneous feedback signals as noisy labels and applying denoising techniques to address them, yet their effectiveness remains limited because erroneous feedback is not random noise by its nature Wang et al. [2021]. Some alternative methods attempt to mitigate erroneous feedback via multi-feedback fusion, yet they often struggle when confronted with conflicting feedback Chen et al. [2021a]. Overall, existing works focus on data mining approaches, lacking a deep understanding of the mismatch between users' implicit feedback and their real preferences.

---

*Corresponding author.

39th Conference on Neural Information Processing Systems (NeurIPS 2025).

In this work, we aim to quantify and uncover users' internal satisfaction with recommendations, thereby bridging the gap between the recommendation model and users' real preferences. In fact, each time user takes an action (e.g., consume an item), a private feeling will be generated in her mind, telling how she is satisfied by taking the action. We summarize such private feeling as *mental reward*. We have following two observations on the mental reward. O1: The mental reward will influence user's short-term interests and her subsequent behaviors. For example, if an user clicked on a news but felt uncomfortable with the content, she would lose interests on that topic and probably not click on similar news again. O2: Users are maximizing their accumulated mental rewards. This is reasonable because users naturally pursue good experiences during interaction with the recommender system. The above observations indicate that mental reward plays an important role in user's sequential decision making. If we can directly optimize user's mental rewards, the recommendation model would be better aligned with users' real preferences.

To this end, we propose MTRec, a novel sequential recommendation framework which uses a learned Mental Reward Model to guide the recommendation model to align with users' real preferences. First of all, we model the user's decision making as a Markov Decision Process (MDP). With the assumption that the user always maximizes her accumulated mental rewards, we use Inverse Reinforcement Learning (IRL) to infer a mental reward function from users' behavioral data. However, plain IRL recovers a deterministic mental reward function, which fails to capture the random nature of the mental rewards. To address this, we propose a Quantile Regression Inverse Q-Learning (QR-IQL) approach to learn a distributional mental reward function, which maps a state-action pair to a distribution of mental rewards. Hence, we use the rewards predicted by the mental reward model as complementary supervision signals to guide the training of recommendation model. In such a way, the misalignment between recommendation model and user's real preferences can be greatly reduced. Experiments on two public datasets show that MTRec significantly improves the performance of several popular recommendation models, in terms of Area Under Curve (AUC) and Normalised Capped Importance Sampling (NCIS). We also test MTRec in Virtual Taobao to demonstrate its effectiveness on reinforcement learning based recommendation models. Moreover, we deployed MTRec in a real-world industrial short video recommendation platform and observed a 7% increase in average user viewing time over a 7-day period during the online A/B test.

Our main contributions are summarized as follows.

- We identify the misalignment problem in sequential recommendation, where erroneous user feedback could severely deviate recommendation model from users' real preferences.

- We introduce MTRec, a novel sequential recommendation framework that aims to bridge the gap between the recommendation model and users' real preferences by a learned mental reward model, which uncovers users' internal satisfaction with recommendations.

- To capture the random nature of the mental rewards, we develop a distributional variant of IRL called QR-IQL to learn the mental reward model. We show how to use the learned mental reward model to guide the optimization of sequential recommendation models.

- We conducted extensive offline and online experiments to demonstrate the improvements brought by MTRec. Additionally, we deployed MTRec in a real-world industrial short video recommendation platform and observed a significant increase in user engagement.

## 2 Related Works

**Implicit user feedback in recommendation.** Since explicit feedback is very sparse, industrial recommender systems rely on implicit feedback (e.g., click, video watching) to train recommendation models Covington et al. [2016], Zhou et al. [2018], Dai et al. [2021], Yang et al. [2021]. However, these models ignore the fact that implicit feedback may not reflect users' real preferences. Some works treat such erroneous feedback as random noise and try to eliminate its influence via denoising, yet these methods have limited accuracy because it is hard to distinguish it from genuine responses Yu and Qin. [2020], Wang et al. [2021]. Another line of works address various types of biases in user feedback, including position bias, selection bias, popularity bias and exposure bias woong Lee et al. [2021], Yi et al. [2023]. However, errors in implicit user feedback differ from the aforementioned biases. In some sense, such errors can be regarded as a certain type of inductive bias Jiawei et al. [2023], because implicit feedback is wrongly assumed to reflect users' real preferences.

**AI alignment through reward model.** Emerging from Natural Language Processing, alignment algorithms have proven effective due to their ability to guide Large Language Models (LLMs) in matching human values. Many popular alignment methods employ a reward model to provide fine-tuning signals. For example, Reinforcement Learning from Human Feedback (RLHF) learns human preferences through a reward model trained with human-rated outputs Ouyang et al. [2022]. Reward rAnked FineTuning (RAFT) uses a reward model to select the best set of training samples based on model outputs Dong et al. [2023]. Inspired by these works, we aim to learn a reward model to guide recommendation models in aligning with users' real preferences. Unlike prior approaches that rely on human annotators to provide reward model labels, our reward model is directly learned from existing user behavioral data, making it more industry-friendly.

**RL and IRL for sequential recommendation.** Sequential recommendation is typically modeled as interactions between users and recommender systems. Prior works have explored using reinforcement learning (RL) to optimize recommendation policy, where the recommender system is modeled as an agent and the users are treated as key components of environment Chen et al. [2019], Zheng et al. [2018]. RL-based methods have great potential to maximize recommender systems' long-term revenue, but they often suffer from the bias of user simulators. Inverse RL (IRL) aims to recover the agent's reward function from expert trajectories. Some works apply IRL to infer the reward function for the recommender system, assuming expert recommendation policies are available Chen et al. [2021b], Liu et al. [2023a]. However, these works ignore that the users are also active agents, and understanding user behaviors is essential for improving recommendation policies. In our work, we model users as agents and use IRL to infer the optimal reward model from their behaviors. Although similarly employing IRL techniques, our work differs fundamentally from existing works by inferring a user-centric reward model, as opposed to system-centric reward modeling.

## 3 Preliminaries

**Sequential recommendation.** Typically, a sequential recommendation model takes a sequence of user-item interactions as input and predicts the next items that mostly attract the users. Due to the lack of explicit feedback, sequential recommendation tasks are usually formulated as predicting the next item that is most likely to induce target user behaviors such as clicks:

$$i_t = \arg\max_{i \in I} p_\phi(\hat{a}|i, h_{t-1}), \tag{1}$$

where $\hat{a}$ represents user's behavior to be predicted, $I = \{i_1, ..., i_N\}$ is the set of candidate items. The parameterized function $p_\phi$ measures the probability of the target behavior $\hat{a}$. The interaction history $h_t$ consists of a list of tuples up to time step $t$, where each tuple $< u, i, a >_t$ consists of a user $u$, an item $i$ and the user's action $a \in A$. In a simplified case, user's action space $A$ can be restricted to $\{click, skip\}$, hence the recommendation task is reduced to predicting the user's click-trough rate.

**Inverse reinforcement learning.** Conventionally, maximum entropy IRL aims to learn a deterministic reward function $r : S \times A \to \mathbb{R}$ by solving Problem 2 Brian et al. [2008].

$$\max_{r \in \mathcal{R}} \min_{\pi \in \Pi} \mathbb{E}_{\pi_E}[r(s, a)] - \mathbb{E}_\pi[r(s, a)] - \mathcal{H}(\pi), \tag{2}$$

where $\mathcal{H}(\pi) \triangleq \mathbb{E}_\pi[-\log \pi(a|s)]$ denotes the entropy of policy $\pi$. Intuitively, this formulation learns a reward function that assigns high reward to the expert policy $\pi_E$ and a low reward to other policies, while searching for the optimal policy under the reward function in the inner loop. The expectation over policies can be replaced by the occupancy measure $\rho_\pi(s, a) = \pi(a|s) \sum_t \gamma^t P(s_t = s|\pi)$, which specifies a probability distribution over $(s, a)$ pairs. Then, Problem 2 can rewritten as:

$$\max_{r \in \mathcal{R}} \min_{\pi \in \Pi} \mathbb{E}_{\rho_E}[r(s, a)] - \mathbb{E}_\rho[r(s, a)] - \mathcal{H}(\pi) - \phi(r), \tag{3}$$

where $\phi$ is a convex regularizer on $r$ Jonathan and Ermon [2016]. If we define the regularizer as $\psi(x) = x - \phi(x)$, then $\mathbb{E}_{\rho_E}[r(s, a)] - \phi(r)$ can be compactly represented by $\mathbb{E}_{\rho_E}[\psi(r(s, a))]$. Moreover, the reward function $r$ can be represented by the soft $Q$-function as

$r(s, a) = Q(s, a) - \gamma \mathbb{E}_{s' \sim P(s,a)} V(s')$, where $V(s) = \mathbb{E}_{a \sim \pi(\cdot|s)}[Q(s, a) - \log \pi(a|s)]$. Also the optimal policy $\pi^*$ can be represented by the soft $Q$-function as $\pi^*(a|s) = \frac{1}{\Delta(s)} \exp(Q(s, a))$, where $\Delta(s) = \sum_{a'} \exp(Q(s, a'))$ is the normalization factor. Then, the inner problem becomes trivial so that Problem 3 can be reformulated as follows Brian et al. [2008], Haarnoja et al. [2018].

$$\max_{Q \in \Omega} \mathbb{E}_{s,a \sim \rho_E}[\psi(Q(s, a) - \gamma \mathbb{E}_{s' \sim P(s,a)} V^*(s'))] - (1 - \gamma) \mathbb{E}_{s_0 \sim \rho_0}[V^*(s_0)], \tag{4}$$

where $V^*(s) = \log \sum_a \exp Q(s, a)$. Note that the objective of Problem 4 depends only on $Q$, which allows us to solve the problem by directly optimizing a Q-network. Solving Problem 4 with the expert data $D_E$ results in the optimal $Q$-function, and the deterministic rewards can be recovered by:

$$r(s, a) = Q(s, a) - \gamma \mathbb{E}_{\rho_E}[\log \sum_a \exp Q(s', a)], \tag{5}$$

**The misalignment problem in recommendation.** In the context of LLM, the alignment problem is defined as optimizing the outputs of LLMs towards matching human values Ouyang et al. [2022]. For recommender systems, the general goal is to maximize users' satisfaction by selecting appropriate items for them. This is naturally in accord with the alignment problem in LLMs, in the sense that aligning with human values is similar to aligning with user preferences. However, directly maximizing user satisfaction is very challenging for recommender systems. Therefore, most existing works focus on optimizing some surrogate objectives such as click-through rate and conversion rate Wang et al. [2017], Zhou et al. [2019]. Although the surrogate objectives make it convenient to optimize recommendation models, they suffer from inductive biases since users' implicit feedback may not reflect their real preferences. As a consequence, *the optimization goal of recommendation models may deviate from users' real preferences.* Compared with one-shot recommendation, the misalignment problem in sequential recommendation is even worse due to the accumulation of errors.

Inspired by recent advances in LLM alignment, we aim to develop a reward model that helps to align recommendation models with users' real preferences. The challenges are two-fold. First, we do not have an off-the-shelf reward model that reflects users' real preferences, under the lack of their explicit feedback. To address this, we propose the Mental Reward model learned from rich user behavioral data to approximate their real preferences. Second, unlike static human values, users' preferences could be highly stochastic, potentially due to unpredictable environmental factors and preference shifts. To resolve this, we propose a distributional Inverse RL approach to capture the randomness of the mental reward model, which will be illustrated in Section 4.

# 4 Method

**Motivation.** User behaviors have been extensively studied in the literature of recommender systems. However, existing works treat user behaviors as labels for recommendation models. In fact, during the interaction with recommender systems, users are active agents rather than static label providers. Moreover, users are implicitly maximizing their own reward functions by taking actions in recommender systems. Based on the above insights, we believe that uncovering the user's reward function would significantly benefit recommendation models in aligning with users' real preferences.

**Overview.** We propose a MenTal reward based Recommendation framework MTRec, which consists of three main parts. First, we introduce a novel User-Centric Markov Decision Process, where uses are modeled as active agents during their interaction with recommender systems. Second, we develop a distributional IRL method called Quantile Regression Inverse Q-learning (QR-IQL) for learning the mental reward model. Last, we show how to use the learned mental reward model to guide existing recommendation models to align with users' real preferences.

## 4.1 User-Centric Markov Decision Process

Reinforcement learning has been applied to model the interaction between users and recommender systems. Existing works usually model the recommender system as agent, who recommends items and receives user feedback as reward Zheng et al. [2018], Chen et al. [2019]. However, they often

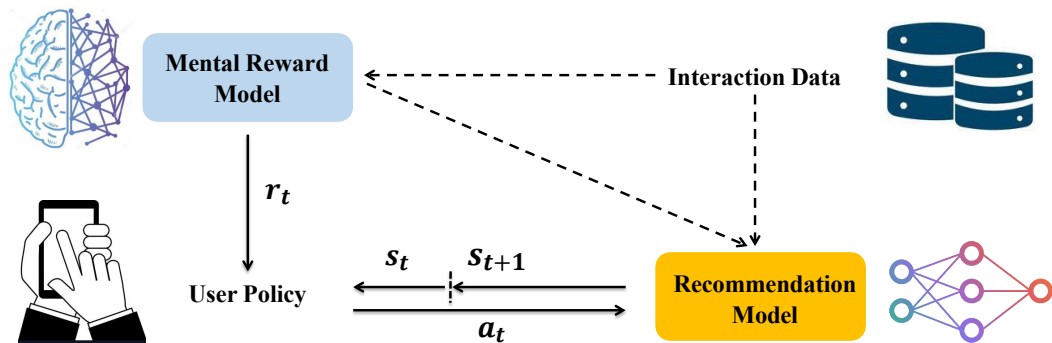

Figure 1: The overall framework of MTRec. The solid lines represent the interaction process. The dashed lines represent the information flow between data and models. Our goal is to recover the mental reward model and use it to improve the recommendation model.

ignore the strategic behavior of the users. In this work, we focus on studying users' behaviors, whose decision making can be modeled as a Markov Decision Process $\mathcal{M} = \langle S, A, P, R, \pi \rangle$:

- $S$ denotes the state space. A state $s_t = (h_t, i_t)$ includes the interaction history up to time $t$. At each time step, the interaction is recorded by a tuple $<u, i, a>_t$, consisting of an user $u$, displayed item $i$ and the user's action $a$.

- $A$ denotes user's action space. An action $a \in \mathcal{A}$ represents user's response to the item displayed to her. User's responses could be of various types, including explicit feedback such as news clicks and video watchings. In this work, we consider a general case where users' responses are either positive or negative, leading to a simplified action space.

- $P : S \times A \rightarrow S$ denotes the transition function. Following existing works on sequential recommendation Zhou et al. [2018, 2019], Wang-Cheng and McAuley [2018], after an user takes an action, a new state $s_{t+1} = s_t \cup \{h_{t+1}, i_{t+1}\}$ will be generated.

- $R : S \times A \rightarrow \theta(r)$ denotes the mental reward function which maps a state and an action to a distribution of mental reward $r \in \mathbb{R}$. Note that we model the mental reward $r$ as a random variable instead of a deterministic value, since $r$ could be influenced by unpredictable environmental factors and preference shifts. Such a modeling allows us to capture the intrinsic randomness and potentially richer information on user's mental rewards.

- $\pi : \mathcal{S} \rightarrow \mu(a)$ denotes the user's behavioral policy, which maps a state to a distribution over actions. A stochastic user policy facilitates reasoning her reward function using IRL.

Figure 1 illustrates the interaction process between user and recommendation model. At each round, the user receives a recommended item and select an action to respond. After that, a mental reward that summarizes user's satisfaction about the current item is generated. Note that the mental reward is highly correlated with user's real preference, but is unknown to the recommendation model. However, the mental reward will influence the user's subsequent behaviors as she seeks to maximize the accumulated mental rewards. We will focus on estimating the mental rewards in the following parts.

## 4.2 Uncovering the Mental Reward Model

Inverse reinforcement learning (IRL) aims to find a reward model that explains behaviors of the expert policy Saurabh and Doshi [2021]. Since the user naturally maximizes her mental rewards, her policy can be regarded as the expert policy. Therefore, IRL could be used to recover the mental reward model using the interaction data. *However, existing IRL methods focus on deterministic reward models, which fail to capture the intrinsic randomness of the user's mental rewards.* To this end, we will develop a distributional version of IRL algorithm for uncovering the mental reward model.

### 4.2.1 A distributional perspective on IRL

Conventional IRL methods such as MaxEntIRL iteratively optimize the reward function and the policy until convergence Brian et al. [2008]. Bayesian IRL methods assume that there are multiple reward

functions and focus on estimating their posterior distribution using Bayes' rule Deepak and Amir [2007], Jaedeug and Kim [2012]. Although Bayesian IRL methods introduce various distributions over reward functions, they are still restricted to deterministic reward functions. By contrast, we aim to learn a distributional reward function where the reward can be stochastic.

Since we model the user's mental reward $r$ as a random variable, Equation 5 can be rephrased using a distributional operator as

$$\mathcal{T}^\pi r(s, a) :\overset{D}{=} Q(s, a) - \gamma \mathbb{E}_{\rho_E}[\log \sum_a \exp Q(s', a)],$$

where $X :\overset{D}{=} U$ denotes equality of probability laws, that is, the random variable $X$ is distributed according to the same law as $U$. As each $Q(s, a)$ function uniquely determines a distribution of $r(s, a)$, learning a distributional reward function is reduced to learning a distributional $Q$-function.

#### 4.2.2 Quantile Regression Inverse Q-learning (QR-IQL)

Problem 4 aims to learn a deterministic $Q$-function, while we aim to learn a distributional $Q$-function. Following the QR-DQN Will et al. [2018], the distribution of $Q$ can be characterized by a quantile distribution. We denote by $Z$ the variable associated with the distribution of $Q$, that is, $Q(s, a) = \mathbb{E}[Z(s, a)]$. Let $\lambda : S \times A \to \mathbb{R}^N$ be a parametric model, where $N$ is the number of quantiles. Then a quantile distribution $Z_\lambda$ maps a state-action pair $(s, a)$ to a uniformly probability distribution supported on $\{\lambda_i(s, a)\}_{i=1}^N$. Instead of learning a scalar value $Q(s, a)$, our model will estimate the positions of supports $\{\lambda_i(s, a)\}_{i=1}^N$ and calculate $Q_\lambda(s, a)$ as:

$$Q_\lambda(s, a) = \frac{1}{N} \sum_{i=1}^N \lambda_i(s, a).$$

Recall that in Problem 4, we optimize a $Q$-network that outputs a scalar $Q$-value. In order to learn the distributional $Q$-function, we change the output layer of the $Q$-network to be of size $|A| \times N$. To derive the objective for the quantile regression inverse Q-learning, we made two modifications on Problem 4. First, since our mental reward model is learned from offline data, the second term in Problem 4 (i.e., $(1 - \gamma)\mathbb{E}_{s_0 \sim \rho_0}[V(s_0)]$) can be replaced by $\mathbb{E}_{(s,a)\sim\rho_E}[V(s) - \gamma V(s')]$. In other words, $\mathbb{E}_{s_0 \sim \rho_0}[V(s_0)]$ is irrelevant with the initial state distribution $\rho_0$. Second, as is suggested in Garg et al. [2021], we choose the regularizer as $\psi(x) = x - \frac{1}{4\alpha}x^2$ for the ease of optimization while bounding the rewards. Finally, we formulate the objective of the quantile regression inverse Q-learning as Problem 6. The complete derivation is provided in Appendix A.1.

$$\max_{Q_\lambda} \mathbb{E}_{\rho_E}[Q_\lambda(s, a) - \log \sum_a \exp Q_\lambda(s, a)] - \frac{1}{4\alpha}\mathbb{E}_{\rho_E}[(Q_\lambda(s, a) - \log \sum_a \exp Q_\lambda(s', a))^2], \quad (6)$$

We apply the Pinball loss to get the optimal quantile distribution supports $\{\lambda_i(s, a)\}_{i=1}^N$ and the optimal $Q_\lambda^*$, which we can use to calculate the mental rewards as:

$$r^*(s, a) = Q_\lambda^*(s, a) - \gamma\mathbb{E}_{\rho_E}[\log \sum_a \exp Q_\lambda^*(s', a)]. \quad (7)$$

Detailed optimization algorithm is provided in Appendix A.2. While building upon ideas from QR-DQN for RL, our QR-IQL is the first distributional IRL algorithm that uncovers the underlying distribution of rewards, which is key to capture the randomness of users' mental rewards.

### 4.3 Applications of the Mental Reward model

Generally, a sequential recommendation model takes user features, candidate item features, interaction histories and some contextual features as input, and output a score for the candidate item used for ranking. For the sake of brevity, we represent the recommendation model as $F_\zeta(i_t|h_t)$, which is parameterized by $\zeta$. Although the mental reward model $r(s, a)$ indicates users' preferences to some extent, it lacks sufficient feature-level modeling and thus cannot be directly used for recommendation. Instead, we use $r(s, a)$ to provide additional learning signals for recommendation models, However, combining $r(s, a)$ with existing recommendation models is non-trivial due to the

variety of learning objectives. Fortunately, these objectives fall into several categories. We will use the following two typical examples to illustrate how to use the mental reward model in practice.

**Classification-based models.** Many sequential recommendation tasks are formulated as binary classification problems with the following Cross Entropy loss:

$$\mathcal{L}_{CE}(\zeta) = -\mathbb{E}_{D_E}[a^P \log(F_\zeta(i|h)) + a^N \log(1 - F_\zeta(i|h))], \tag{8}$$

where $a^P = 1$ and $a^N = 0$ indicate user's positive and negative responses, and $F_\zeta(i|h)$ represents the estimated probability of clicking on item $i$ based on history $h$. We want the recommendation models to also maximize the expectation of user's mental rewards, leading to the following alignment loss:

$$\mathcal{L}_{Align}(\zeta) = -\mathbb{E}_{D_E}[r^*(s, a^P) \cdot F_\zeta(i|h) + r^*(s, a^N) \cdot (1 - F_\zeta(i|h))], \tag{9}$$

Then, the final loss for training the recommendation model can be written as a weighted combination of the two losses:

$$\mathcal{L}_{Final}(\zeta) = \mathcal{L}_{CE}(\zeta) + \kappa \cdot \mathcal{L}_{Align}(\zeta). \tag{10}$$

**RL-based models.** In a typical setting, the recommender system is modeled as an agent, who maximizes the accumulated system rewards. We denote by $\hat{r}(h, i, a)$ the system reward after recommending item $i$ and receiving user feedback $a$ given interaction history $h$. In this context, $F_\zeta(i|h)$ represents the RL-based recommendation policy, whose goal is to maximize the expectation of accumulated rewards $\hat{r}$:

$$\mathcal{L}_{RL}(\zeta) = -\mathbb{E}_{i_t \sim F_\zeta}[\sum_t \hat{r}(h_{t-1}, i_t, a_t)] \tag{11}$$

Since we want the recommendation model to also maximize the mental rewards of the user. We simply add the mental reward $r^*$ to $\hat{r}$ and obtain the following objective.

$$\mathcal{L}_{Final}(\zeta) = -\mathbb{E}_{i_t \sim F_\zeta}[\sum_t \hat{r}(h_{t-1}, i_t, a_t) + \kappa \cdot r^*(h_{t-1}, i_t, a_t)] \tag{12}$$

See Appendix A.3 for more implementation details.

# 5 Experiments

In this section, we report the performance of MTRec in both offline and online settings, with focuses on answering the following research questions (RQs).

- (RQ1:) How does MTRec improve classification-based recommendation models?
- (RQ2:) How does MTRec improve RL-based recommendation models?
- (RQ3:) Does the learned mental reward model provide useful information?
- (RQ4:) How does MTRec perform in online A/B test?

## 5.1 Experiments on Public Datasets (RQ1)

**Datasets.** The Amazon dataset McAuley et al. [2015] collects user review data from Amazon e-commerce platform. We use two subsets of the Amazon dataset: Books and Electronics in our offline experiments. More details on processing the datasets are provided in Appendix A.4.

**Baselines.** We use eight widely used recommendation models as baselines and combine each of them with MTRec to test the improvements brought by MTRec. **Wide&Deep** Cheng et al. [2016] is a hybrid recommendation model combining a wide linear model and deep neural network for collaborative filtering. **PNN** Qu et al. [2016] is a neural network architecture designed for CTR prediction in recommender systems. **DeepFM** Guo et al. [2017] is a hybrid recommendation model combining factorization machines and deep neural networks. **SASRec** Wang-Cheng and McAuley [2018] uses the self-attention mechanism to model sequential patterns for recommendation systems. **DIN** Zhou et al. [2018] is an attention-based neural model for sequential recommendation, where the attention mechanism aims to distinguish the interest of a user's historical behaviors. **DIEN** Zhou et al. [2019] designs a sequential architecture to model interest evolution for recommendation,

which uses an auxiliary loss to capture temporal interests. **LinRec** Liu et al. [2023b] is a lightweight linear recommendation model designed for efficient computation and scalability with large datasets. **SIGMA** Liu et al. [2024] is a sequential recommendation model that uses a selective gating mechanism to focus on the most relevant user behaviors for improved performance.

**Evaluation metrics.** We use the following two metrics: Area Under Curve (AUC) Fawcett [2006] and Normalised Capped Importance Sampling (NCIS) Swaminathan and Joachims [2015]. AUC is used to measure the model's ranking ability and NCIS is used to approximate the model's online performance Gilotte et al. [2018]. Formally, the score of NCIS can be calculated by:

$$\tilde{J}^{NCIS}(\mathcal{M}) = \frac{\sum_{i=1}^{n} \tilde{\rho}_i(\mathcal{M}) * L_i}{\sum_{i=1}^{n} \tilde{\rho}_i(\mathcal{M})},$$

where $\rho_i(\mathcal{M}) = \prod_{t \in \mathcal{T}} p_t(\mathcal{M})$ is the probability that the CTR model $\mathcal{M}$ follows the request trajectory of the user $i$, $p_t(\mathcal{M})$ is the click-through rate estimate of model $\mathcal{M}$ for item $t$ and $n$ is the number of users in the test set for NCIS. In the experiments, we use the complete trajectories of 10% of users to calculate NCIS. Moreover, we obtain the final NCIS score by substracting the NCIS of the untrained model from that of the trained model to eliminate the impact of random parameters among different models. Intuitively, $\tilde{J}^{NCIS}(\mathcal{M})$ awards a CTR model with a high score if the model has large probability to follow long trajectories.

**Results on Amazon datasets.** Table 1 summarizes the experimental results on the two Amazon datasets. The suffix "IRL" indicates that the mental reward model is learned by a non-distributional IRL algorithm IQ-Learn Garg et al. [2021], while the suffix "MTRec" indicates that the mental reward model is learned by our algorithm QR-IQL. It can be observed that integrating MTRec with existing models consistently improves their AUC and NCIS across almost all baseline models. Moreover, the improvements on NCIS are generally more significant than AUC. This aligns with the motivation of MTRec, that is, to maximize overall user satisfaction and long-term engagement. Note that the AUC is also slightly improved, demonstrating that the improvement of users' long-term engagement is not at the sacrifice of the model's ranking ability. In addition, the comparisons between IRL and MTRec demonstrates the benefit of learning a distributional version of mental reward model.

| Model | Electronics | | Books | |
|---|---|---|---|---|
| | AUC | NCIS | AUC | NCIS |
| Wide&Deep | 0.8290 | 0.6749 | 0.8605 | 2.0967 |
| Wide&Deep-IRL | 0.8342 | 0.8738 | **0.8661** | 3.1264 |
| Wide&Deep-MTRec | **0.8351** | **0.9063** | 0.8657 | **3.2043** |
| PNN | 0.8396 | 0.6618 | 0.8603 | 2.3316 |
| PNN-IRL | **0.8547** | 0.9297 | 0.8673 | 3.4567 |
| PNN-MTRec | 0.8542 | **0.9515** | **0.8679** | **3.4971** |
| DeepFM | 0.8424 | 0.6993 | 0.8634 | 2.3146 |
| DeepFM-IRL | 0.8458 | 0.8542 | 0.8715 | 3.7296 |
| DeepFM-MTRec | **0.8468** | **0.8961** | **0.8742** | **3.7904** |
| SASRec | 0.8325 | 0.8243 | 0.8675 | 3.1755 |
| SASRec-IRL | **0.8366** | 0.8681 | 0.8681 | 3.1946 |
| SASRec-MTRec | 0.8328 | **0.8833** | **0.8798** | **3.4634** |
| DIN | 0.8523 | 0.6044 | 0.8653 | 2.1324 |
| DIN-IRL | 0.8533 | 0.8168 | 0.8701 | 3.1675 |
| DIN-MTRec | **0.8542** | **0.8728** | **0.8732** | **3.2208** |
| DIEN | 0.8448 | 0.7766 | 0.8686 | 2.2685 |
| DIEN-IRL | 0.8461 | 0.8857 | 0.8723 | 3.0476 |
| DIEN-MTRec | **0.8472** | **0.9324** | **0.8757** | **3.1185** |
| LinRec | 0.8579 | 0.8077 | 0.8754 | 2.4653 |
| LinRec-IRL | **0.8597** | 0.9365 | 0.8771 | **3.8005** |
| LinRec-MTRec | 0.8594 | **0.9782** | **0.8792** | 3.7928 |
| SIGMA | 0.8581 | 0.7946 | 0.8762 | 2.3975 |
| SIGMA-IRL | 0.8592 | 0.9261 | 0.8802 | 3.7556 |
| SIGMA-MTRec | **0.8604** | **0.9563** | **0.8814** | **3.8025** |

Table 1: Experimental results on Amazon datasets.

### 5.2 Experiments on Virtual Taobao (RQ2)

In this set of experiments, we choose RL-based recommendation models as our baselines and combine them with MTRec to test their performance. Both training and testing of the algorithms are conducted in simulated interactive recommendation environments on Virtual Taobao Shi et al. [2019]. We construct an expert dataset containing 100,000 high-quality trajectories by recording trajectories with averaged CTR>0.5. After training the mental reward model using the expert data, we add the predicted mental rewards to the original rewards and train the baselines again.

**Baselines.** Virtual Taobao allows training recommendation policies by RL. We use Proximal Policy Optimization (PPO) Schulman et al. [2017] and Soft Actor-Critic (SAC) Haarnoja et al. [2018] as baselines. According to Equation 12, RL-based recommendation models can be adjusted by simply adding the mental rewards to the base rewards $r_{final} = r_{env} + \kappa \cdot r_{mental}$. In Virtual Taobao, $r_{env}$ is provided by a pre-trained user model. We set $\kappa = 0.2$ for trade-off between the two reward signals.

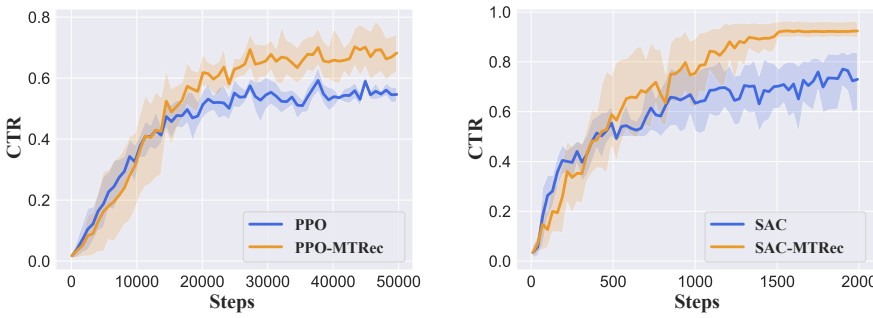

Figure 2: Training curves of RL models. Averaged CTR is reported with 95% confidence interval.

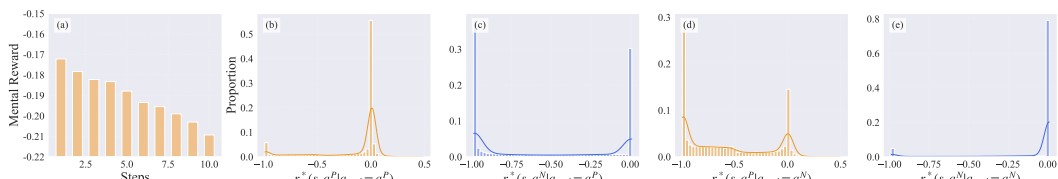

Figure 3: Illustrations of the predicted mental rewards. (a) Averaged mental rewards by steps in all trajectories; (b-e) Expected and counterfactual mental rewards given actual user actions.

**Evaluation metric.** The major evaluation metric used in Virtual Taobao is the episodic Click-Through-Rate (eCTR) during the simulated online interaction. The eCTR is calculated as:

$$\text{eCTR} = \frac{r_{episode}}{10 * N_{step}},$$

where 10 is the number of items recommended in a single page, $r_{episode}$ is the total number of clicks in an episode and $N_{step}$ is the total number of steps.

**Results on Virtual Taobao.** From Fig 2 we can see that the baselines attain average CTRs of 0.5435 and 0.7055 respectively. By incorporating the mental rewards during training, PPO and SAC show significantly improved performance at average CTRs of 0.678 and 0.909 respectively. This demonstrates that the mental rewards provide more useful information about users' real preferences and successfully boost the performance of RL-based recommendation models.

### 5.3 Evaluation of the Mental Reward Model (RQ3)

It is almost impossible to evaluate the learned mental reward model directly due to the lack of ground-truth mental reward labels. Yet, we evaluate the mental reward model indirectly by its correlation with the datasets. First, we calculate the averaged mental rewards at different steps in all trajectories. Figure 3 (a) shows the results of the first ten steps on Amazon Book dataset. We can see that the averaged mental rewards decrease obviously with the increase of steps. This result aligns with our intuition that users may get tired and receive less mental rewards during their interaction with the recommender system.

Second, we visualize the distribution of the mental rewards conditioned on the users' real responses. For instance, Figure 3 (b) and (c) show the distributions of expected mental reward $r^*(s, a^P|a_{real} = a^P)$ and the counterfactual mental reward $r^*(s, a^N|a_{real} = a^P)$ when the users take a positive action $a_{real} = a^P$. Specifically, $r^*(s, a^P|a_{real} = a^P)$ represents the predicted user's mental reward after she takes action $a^P$, while $r^*(s, a^N|a_{real} = a^P)$ represents the counterfactual mental reward if she had taken a negative action $a^N$. Intuitively, an user should receive a relatively high reward for the action she actually taken (a high $r^*(s, a_{real}|a_{real})$). Yet, the experimental results show that a non-negligible proportion of $r^*(s, a_{real}|a_{real})$ is relatively low, which suggests that there is a mismatch between users' actions and their real preferences (recall the example in the abstract: a user

might click on a news article because of its attractive headline, but end up feeling uncomfortable after reading it). These observations indirectly validate the effectiveness of the mental reward model.

### 5.4 Online A/B test on Industrial Platform (RQ4)

To further validate the effectiveness of MTRec, we deploy it on an industrial short-video recommendation platform, which has tens of millions of Daily Active Users (DAU). Short-video recommendation is a typical sequential recommendation scenario, where the goal is to improve the user engagement with the platform. The baseline model is a Deep Cross Network (DCN) Wang et al. [2017], which is trained using binary labels indicating whether the users click on videos. However, intuitively, clicking on a video does not necessarily mean that the user is satisfied after watching the video. Therefore, we expect that MTRec could help to improve the overall users' satisfaction on the recommended videos and hence improve their engagement.

We improve the DCN model by incorporating the alignment loss, as illustrated in Equation 10. We find that the averaged video viewing time stably improve by about 7% compared with the baseline model, which demonstrates that MTRec indeed improves the overall recommendation quality and leads to better user engagement. Note that MTRec is quite industrial friendly because of two reasons: (1) the dataset used to train MTRec could be same with that used to train the recommendation model; (2) we only need to add an auxiliary alignment loss to the original recommendation loss.

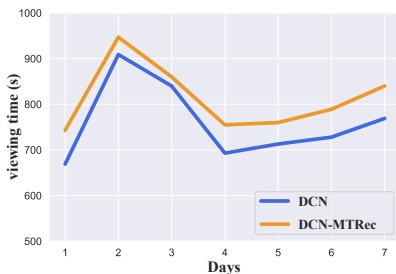

Figure 4: Online A/B test results.

## 6 Conclusions

The general goal of recommender systems is to satisfy users by providing items that align with their real preferences. However, existing works focus on optimizing surrogate objectives based on users' implicit feedback, ignoring that the implicit feedback may not accurately reflect their real preferences. Consequently, recommendation models could be systematically biased. In this work, we aim to fill this gap by studying the users' behaviors and uncovering the distribution of their mental rewards. We propose a novel distributional IRL algorithm to learn the mental reward model and use it to guide the training of recommendation models. Finally, we validate the effectiveness of MTRec via both offline and online experiments, including the A/B on an industrial short video recommendation platform. In the end, we believe that existing studies on user modeling are far from totally understanding users.

## Acknowledgements

This research is supported by Guangdong Basic and Applied Basic Research Foundation (2025A1515010247), and the Fundamental Research Funds for the Central Universities (2024ZYGXZR069).

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

## A    Technical Appendices and Supplementary Material

### A.1    Derivations of Problem 6

In this section, we show that Problem 4 can be simplified and translated to Problem 6, under the assumption that $Q(s, a)$ follows a quantile distribution $Z_\lambda(s, a)$. Given Problem 4:

$$\max_{Q \in \Omega} \mathbb{E}_{s, a \sim \rho_E}[\psi(Q(s, a) - \gamma \mathbb{E}_{s' \sim P(s, a)} V^*(s'))] - (1 - \gamma) \mathbb{E}_{s_0 \sim \rho_0}[V^*(s_0)], \qquad (4)$$

The second term can be expanded as:

$$(1 - \gamma) \mathbb{E}_{s_0 \sim \rho_0}[V^*(s_0)] = (1 - \gamma) \sum_{t=0}^{\infty} \mathbb{E}_{D_E}[V^*(s_t)] - (1 - \gamma) \sum_{t=1}^{\infty} \mathbb{E}_{D_E}[V^*(s_t)]$$
$$= \mathbb{E}_{\rho_E}[V^*(s) - \gamma V^*(s')], \qquad (13)$$

Since we already have the expert trajectory dataset $D_E$, we will use $\rho_E$ to estimate Equation 13. Then, on substituting $\psi(x) = x - \frac{1}{4\alpha}x^2$ and $V^*(s) = \log \sum_a \exp Q(s, a)$ in Problem 4, we have:

$$\max_Q \mathbb{E}_{\rho_E}[Q(s, a) - \log \sum_a \exp Q(s, a)] - \frac{1}{4\alpha} \mathbb{E}_{\rho_E}[(Q(s, a) - \log \sum_a \exp Q(s', a))^2]. \qquad (14)$$

As $Q(s, a)$ is parameterized by quantiles $\{\lambda_i(s, a)\}_{i=1}^N$, we replace $Q$ with $Q_\lambda$ and obtain Problem 6.

## A.2 Optimization steps for QR-IQL

In order to learn the distributional $Q$-function, we change the output layer of the $Q$-network to be of size $|A| \times N$, where $|A|$ denotes the size of the action space and $N$ denotes the number of quantiles. For a given action, each of the $N$ heads implicitly correlates to a $\lambda_i$. We adopt the widely used Pinball loss to learn the positions of $\{\lambda_i\}_{i=1}^N$, which is defined as:

$$p_\lambda(u) = \begin{cases} \lambda \cdot u, & u \geq 0 \\ (\lambda - 1) \cdot u, & u < 0 \end{cases}$$

where $u$ represents the error between the predicted value and the target value at quantile $\lambda$. Intuitively, the Pinball loss pushes the quantile $\lambda$ to the right position so that the predicted value distribution matches the target distribution. Based on the objective of Problem 6, we define two errors as:

$$u^1_{\lambda_i}(s, a) = \lambda_i(s, a) - \log \sum_a \exp \lambda_i(s, a),$$

$$u^2_{\lambda_i}(s, a, s') = \frac{1}{4\alpha}(\lambda_i(s, a) - \log \sum_a \exp \lambda_i(s', a))^2,$$

where $\lambda_i$ is the $i$-th quantile and $\lambda_i(s, a)$ is the corresponding head of the $Q_\lambda$-network. Complete training procedures are provided in Algorithm 1.

---

**Algorithm 1** QR-IQL Optimization Steps

---

**Input**: Interaction (expert) data $D_E$, number of quantiles $N$
1: Initialize network $Q_\lambda$;
2: **repeat**
3:     Sample (batched) data $(s, a, s')$;
4:     Compute errors $u^1_{\lambda_i}(s, a)$ and $u^2_{\lambda_i}(s, a, s')$ for each quantile in $\{\lambda_i\}_{i=1}^N$;
5:     Compute the Pinball losses $p_{\lambda_i}(u^1)$ and $p_{\lambda_i}(u^2)$ for each quantile in $\{\lambda_i\}_{i=1}^N$;
6:     Compute the total loss as: $\sum_{i=1}^N [p_{\lambda_i}(u^1) + p_{\lambda_i}(u^2)]$;
7:     Minimize the total loss by Adam Kingma and Ba [2015];
8: **until** convergence
**Output**: $Q_\lambda^*$

---

## A.3 Implementation Details

Our experiments are run on a server with 2×AMD EPYC 7542 32-Core Processor CPU and 2×NVIDIA RTX 3090 graphics. For the offline experiments on Amazon datasets, it takes about 3 hours for 50,000 iterations of training with a 4000 batch size. For online experiments on Virtual Taobao, it takes about 4 hours for 50,000 RL training steps.

Algorithm 2 describes an overview of the implementation procedures. Basically, there are two stages. At stage 1 we focus on learning $Q_\lambda^*$ and at stage 2 we focus on learning $F_\zeta^*$. In practice, the architecture of the recommendation model $F_\zeta$ could be of various types. For example, in Section 5.1, we test seven widely used recommendation models: Wide&Deep Cheng et al. [2016], PNN Qu et al. [2016], DeepFM Guo et al. [2017], DIN Zhou et al. [2018], DIEN Zhou et al. [2019], LinRec Liu et al. [2023b] and SIGMA Liu et al. [2024]. We will use the same network architecture of $F_\zeta$ to construct $Q_\lambda$ (except the output layer) to ensure that the features are processed properly. All the hyper-parameters of the backbone models follow their official codes. For implementation of MTRec, we select the number of quantiles $N = 10$ and the weight $\alpha = 0.5$ in Problem 6.

Actually, our MTRec framework is industrial friendly due to the following merits. First, all the training procedures are run in an offline manner, saving the cost of building online user-system interaction environments. Second, since $Q_\lambda$ shares the model architecture with $F_\zeta$, we do not need to build a mental reward model from scratch, saving a lot of work for adaptation.

## A.4 Details on the Amazon datasets

The Amazon dataset McAuley et al. [2015] collects user review data from amazon.com. The crawled reviews have a time span from May 1996 to July 2014. The dataset can be divided into many subsets

---
**Algorithm 2** Overall Implementation of MTRec
---
**Input**: Interaction (expert) data $D_E$

  1: Initialize networks $Q_\lambda$ and $F_\zeta$;
  2: Learn $Q_\lambda^*$ according to Algorithm 1;
  3: Obtain the mental rewards by
     $r(s, a) \leftarrow Q_\lambda^*(s, a) - \gamma V^*(s'), \forall s, a \sim D_E$;
  4: Patch the mental rewards to $D_E$;
  5: Learn $F_\zeta^*$ according to Equation 10 or Equation 12;

**Output**: Aligned Recommendation Model $F_\zeta^*$
---

according to the various product categories. To verify the effectiveness of MTRec, we utilized two subsets of the Amazon dataset: Books and Electronics. We treated the reviews as user behaviors and sorted the reviews from each user chronologically. Based on a user's historical behaviors, our goal was to predict whether the user would write a review.

| Dataset | User | Item | Categoriy | Sample |
|---|---|---|---|---|
| Books | 603,668 | 367,982 | 1,600 | 603,668 |
| Electronics | 192,403 | 63,001 | 801 | 192,403 |

Table 2: The statistics of the Amazon datasets.

## A.5   Limitations

While our work opens a door for studying users' intrinsic rewards during their interaction with the recommender systems, it still lacks a systematic method to thoroughly evaluate the learned mental reward model. In future works, we plan to construct a comprehensive benchmark involving large-scale human studies to further evaluate the mental reward model.

