# OpenReview forum: "MTRec: Learning to Align with User Preferences via Mental Reward Models"
_NeurIPS.cc/2025/Conference — NeurIPS 2025 poster_

### Official Review · Reviewer_nSej · 2025-06-28

**Clarity:** 4
**Significance:** 3
**Originality:** 3
**Rating:** 5
**Confidence:** 4

**Summary:**

This paper introduces a mechanism to align recommendation systems with user preferences by modeling a user as an agent with their own mental reward function, which is used to provide a learning signal to two classes of recommendation systems: (1) classification models and (2) RL agents. The authors introduce an algorithm to learn mental reward function using a novel quantile-based inverse RL to learn a quantile Q function, which is then used to produce the reward value. Their evaluations included performance on standard recommendation system benchmarks, and end-to-end performance from the real video recommendation system. The results suggest improvement over the baseline methods.

**Questions:**

Thank you for your work on this paper. I enjoyed reading it.

* I’d suggest including more discussion about mental reward function, and its connection to other sciences. This can help validate this concept and make the result stronger.
* I’d suggest addressing minor comments:
  * Adding standard deviation to Figure 3A
  * Adding missing citations (see my comment in Strengths and Weaknesses)

**Ethical Concerns:**

["NO or VERY MINOR ethics concerns only"]

**Final Justification:**

This was a strong paper with novel contributions about mental reward model. I described its strength and weaknesses in my original review. Authors clarified connection of mental reward models to psychology, especially the Likert evaluation methods. This clarified the novelty of this work, which is why I raised my score.

**Limitations:**

Yes

**Quality:**

3

**Strengths And Weaknesses:**

# Quality

* (+) A nice paper that provides mathematical rigor with practical algorithms.
* (+) Good discussions about interpretability of the mental reward function.

# Clarity
* (+) Clear description of research questions and their evaluation.
* (+) Math derivations are clear to follow.
* (-) Missing citations for some claims made in the paper.
  * “User behaviors have been extensively studied in the literature of recommender systems”
* (-) Figure 3a doesn’t include the standard deviation. It would be useful to see how noisy the measurements are for each step.

# Significance

* (+) This work provides practical algorithms to improve recommendation systems. The quantile IRL algorithm learns from offline data, which makes it easier to adopt.

# Originality

* (+) Authors model users as RL agents, where the reward function is modeled as a stochastic function, which previous approaches didn’t do (prev. used are deterministic reward functions).
* (-) It wasn’t clear to me what inspired the mental reward function. It seems like it can be related to some concepts in psychology (i.e. dopamine). I’d like to see some discussion and potential connections to other sciences.

---

> ### Author Rebuttal · Authors · 2025-07-27
>
> Thanks for your appreciation and useful suggestions to our work. Please see our responses below.
>
> **Q1: How does the concept of mental reward relate to other sciences?**
>
> **A1:** Thanks for raising this interesting question. To be honest, we don't have a clear answer for now. But we would like to share some of our thoughts on this. First, our study is definitely related to psychology, and perhaps biology. In psychology, if we want to evaluate a person's preference on an item, we usually ask a set of questions from multiple perspectives (e.g., Likert Scale) and build some models (e.g., regression) to analyze the person's feedback. However, in the recommendation process, we don't have the chance to ask user many questions about their feelings on an item, therefore the aforementioned psychology models may not work here. But perhaps there are other psychology models that focus on estimating user's preferences from very limited feedback. We will look into this in future works.
>
>
> **Q2: Adding standard deviation to Figure 3A.**
>
> **A2:** This is a very helpful suggestion! Unfortunately, we can't upload figures in the rebuttal phase according to NeurIPS's policy.  We will update the figure in the next version of our paper.
>
>
> **Q3: Adding more citations on user behavior modeling in recommender systems.**
>
> **A3:** Sure, we will add those citations and check other parts of the paper. Thanks for your advice!
>
> Thanks again for your time spent on our paper. We would be very grateful if you could re-evaluate our work based on our rebuttal.

---

> ### Author Response · Authors · 2025-08-06
> **A gentle reminder for author-reviewer discussions**
>
> Dea Reviewer nSej,
>
> Thank you for your insightful feedback and appreciation to our work. Your kind suggestions are really helpful and shed light on the future directions of our work. As the deadline of the author-reviewer discussion phase approaching, could you please let us know if there are any remaining concerns or anything we can do to further improve our work?
>
> Warm regards,
>
> The authors.

---

> > ### Comment · Reviewer_nSej · 2025-08-08
> > **Response to the rebuttal #1**
> >
> > Hello,
> >
> > Thank you for the thoughtful response to my comments. Indeed, your answer to Q1 has clarified my question. I think it's worthwhile to add this discussion to the paper, especially comparison to Likert surveys. It helps to clarify contribution of mental reward models. I will raise my originality and clarify scores.
> >
> > I sincerely apologize for the delay in my feedback! I hope this didn't cause you too much stress.

---

> > > ### Author Response · Authors · 2025-08-08
> > > **Thank you Reviewer nSej**
> > >
> > > Dear Reviewer nSej,
> > >
> > > We are very glad to hear from you. We will definitely add the thoughtful discussions with you to the next version of the paper. Your kind suggestions really make this work better.
> > >
> > > Meanwhile, as your main concerns in the review have been addressed, would you consider raising the overall score to 5？That would be a great support to us. We will also fully respect that if you decide to keep the current rating.
> > >
> > > Best wishes,
> > >
> > > The authors.

---

### Official Review · Reviewer_bFhU · 2025-07-03

**Clarity:** 3
**Significance:** 2
**Originality:** 3
**Rating:** 4
**Confidence:** 3

**Summary:**

The paper address the critical issue that implicit feedback often misrepresents user satisfaction. Then, the authors proposes MTRec, a sequential recommendation framework that leverages a mental reward model learned via distributional inverse reinforcement learning (IRL) to align recommendation models with users’ real preferences. Experiments on public datasets and an industrial platform demonstrate improvements.

**Questions:**

None

**Ethical Concerns:**

["NO or VERY MINOR ethics concerns only"]

**Final Justification:**

The difficulty of achieving a well-defined mental reward remains and the effort on the extended ablation in the rebuttal partially addressed my concern. Though the concerns in the discussion are only partially addressed in my view and there is still room to improve the convincingness of the paper, I believe the current revised state of the work is reaching the bar of NeurIPS so I am raising the rating to 4.

**Limitations:**

Light

**Quality:**

2

**Strengths And Weaknesses:**

Strengths:
* Novel Problem Formulation: Addresses the fundamental mismatch between implicit feedback and user satisfaction, a long-standing issue in recommendation systems.
* Technical Innovation: The proposed QR-IQL algorithm, a distributional variant of IRL, effectively captures the stochasticity of user satisfaction—a significant advancement over deterministic reward models.
* Practical Impact: The industrial deployment of MTRec in  real-world and achieve good improvment.

Weaknesses:
* Mental Reward Interpretation: While the paper argues that mental rewards reflect user satisfaction, the model’s direct interpretability is limited. The absence of ground-truth mental reward labels necessitates indirect validation (e.g., correlation with user actions), which leaves open questions about how accurately the model captures true preferences.
* Oversimplified Modeling of User Feedback: The framework assumes user mental rewards can be modeled via distributional IRL, but real-world feedback exhibits fragmentation and uncertainty that may limit model accuracy.(For instance, a user might skip a video due to network issues or just  bad mood).
* Inadequate Experiments: The paper only gives main experiments and online experiments, but lack of ablation experiments to prove their performance of method. (for example: Hyperparameter Selection(different alignment loss weights and number of quantiles N), quantile regression vs. distributional rewards , Computational cost....)

---

> ### Author Rebuttal · Authors · 2025-07-30
>
> Thanks for your valuable suggestions to our work. Please see our responses below.
>
> **Q1: The model’s direct interpretation on the mental reward is limited.**
>
> **A1:** Thanks for raising this question. In the third paragraph of the Introduction, we define the user's mental reward as "the summary of her private feelings about how she is satisfied". We also use multiple examples to illustrate this concept. However, we agree with you that we did not provide a very specific definition of the mental reward, because it is essentially **a highly abstract concept and can be interpreted in different ways depending on the scenario contexts**. For example, a user might click on a news article because of its attractive headline, but end up feeling uncomfortable after reading the content. Here, the mental reward is a reflection and quantification of how the user feels uncomfortable.
>
> On the other hand, despite its definition, the mental reward **does exist**, and it lies between user's implicit feedback (e.g., click) and her real preferences. Our work aims to narrow this gap by using the mental reward as an effective tool (and the experimental results demonstrate this). In summary, our work makes an important step towards understanding the gap between users' implicit feedback and their real preference. We contribute a novel and theoretically solid algorithm to uncover the mental rewards and demonstrate its effectiveness by extensive experimental results.
>
> In addition, we agree with the reviewer that evaluation about how accurately the model captures true preferences is an open question. In our paper, we compute the **correlation between the mental rewards and the lengths of user-RS interaction trajectories**. We also **visualize the distribution of the mental rewards conditioned on the users’ real actions** (please refer to Section 5.3). We believe that these results are instructive in answering the above question.
>
> **Q2: Oversimplified Modeling of User Feedback.**
>
> **A2:** Thanks for mentioning this issue. Real-world recommender systems may involve many types of user feedback, depending on specific application scenarios. For research works, we have to make appropriate simplification. In fact, we follow existing literature (see related works and Section 2 and 5) and classify user feedback as positive and negative, in order to cover as many scenarios as possible. However, we aggree that extending MTRec to include more diverse user feedback is an interesting future direction.
>
> In addition, the reviewer mentioned that the uncertainty in user feedback may influence the accuracy of the mental reward model. This is true. Actually, such uncertainty affects almost all the recommendation models, because **user feedback is random variable by its nature**. To mitigate this issue, we model the mental reward also as random variable, which motivates the design of our algorithm QR-IQL (see Secion 4.2).  QR-IQL addresses the uncertainty in the mental rewards and achieves better performace than conventional IRL algorithm (please refer to Section 5.1).
>
> **Q3: More ablation experiments.**
>
> **A3:** We agree that including more experimental results would strengthen our claims. Therefore, we add three new set of experiments to test different choices of hyper-parameters. The results are shown below.
>
> **Exp1: Ablation on different choices of N (number of quantiles)**. Intuitively, a larger N (more quantiles) provides more flexibility to model the underlying distribution of $Q_\lambda$, however, it may also bring overfitting issues. We agree that searching for the optimal N is necessary. Therefore, we add a new set of experiments on the Electronic dataset over three models, as shown below.
>
> |    Model    | AUC (N=5)| AUC (N=10)| AUC (N=15)| AUC (N=20)| NCIS (N=5)| NCIS (N=10)| NCIS (N=15)| NCIS (N=20)|
> | :----------- | :-----------: | :-----------: | :-----------: | :-----------: | :-----------: | :-----------: | :-----------: | :-----------: |
> | DeepFM-MTRec      | 0.8462      |**0.8468**    | 0.8467       | 0.8463      | 0.8960      |0.8961      | 0.8960       |**0.8975**      |
> |DIN-MTRec               | 0.8544      |0.8542     | 0.8544      | **0.8547**      | 0.8722      | 0.8728     | **0.8733**       |0.8731      |
> | LinRec-MTRec         | 0.8593      |0.8594     | **0.8597**      | 0.8589      | 0.9773      |0.9782      | 0.9780       |**0.9785**      |
>
> From the above table, we can see that different choices of N do not result in significant changes in terms of AUC and NCIS.  In addition, N=5 performs the worst, perhaps due to limited flexibility to represent the underlying quantile distribution.
>
> We also test different choices of N on Virtual Taobal over two RL-based methods. The results are reported below.
>
> |    Model    | Averaged CTR (N=5)| Averaged CTR (N=10)| Averaged CTR (N=15)| Averaged CTR (N=20)|
> | :----------- | :-----------: | :-----------: | :-----------: | :-----------: |
> | PPO-MTRec      | 0.665      |0.678   | 0.686       | **0.688**      |
> |SAC-MTRec               | 0.885      |0.909     | 0.915      | **0.917**      |
>
> Surprisingly, we obverve a clear trend of improving performance, with respect to N, of both PPO and SAC. These results show that a larger number of quantiles helps to learn more accurate mental reward models. Note that this trend is not that obvious in supervised learning models (refer to the first table). We hypothesis that it is because RL-based methods are more sensitive to rewards.
>
> **Exp2: Ablation on different choices alignment loss weights $\kappa$**. As RL-based methods are more sensitive to rewards, we test 7 different choices of $\kappa$ under the PPO and SAC experiments. The results are shown below.
> |    Model    | Averaged CTR ($\kappa$=0.1)| Averaged CTR ($\kappa$=0.2)| Averaged CTR($\kappa$=0.3)| Averaged CTR ($\kappa$=0.4)| Averaged CTR ($\kappa$=0.5)| Averaged CTR ($\kappa$=0.6)| Averaged CTR ($\kappa$=0.7)|
> | :----------- | :-----------: | :-----------: | :-----------: | :-----------: |:-----------: | :-----------: | :-----------: |
> | PPO-MTRec      | 0.698      |0.678   | 0.706         | **0.718**      | 0.703   | 0.693       | 0.704     |
> |SAC-MTRec        | 0.892      |0.909     | **0.927**   | 0.917      | 0.904   | 0.915       | 0.896      |
>
> We can see that the optimal choices of $\kappa$ are 0.4 and 0.3 respectively. We can see that increasing $\kappa$ does not always bring improvement. This indicates that the mental reward is better used as regularizer, instead of the main source of the reward. We hypothesis that the mental rewards, while provide useful supervision information, may also bring inductive bias.
>
> **Exp3: Computational costs**.
> Our experiments are run on a server with 2×AMD EPYC 7542 32-Core Processor CPU and 2×NVIDIA RTX 3090 graphics. For the offline experiments on Amazon datasets, it takes about 3 hours for 50,000 iterations of training with a 4000 batch size. For online experiments on Virtual Taobao, it takes about 4 hours for 50,000 RL training steps.
>
> Furthermore, although training MTRec's additional mental reward model takes several hours, this cost is worthwhile since recommender systems are primarily concerned with inference latency, and MTRec adds no computation during inference.
>
> Thanks again for your time spent on our paper. We would be very grateful if you could re-evaluate our work based on our rebuttal.

---

> > ### Comment · Reviewer_bFhU · 2025-08-06
> > **Reviewer bFhU's Comment on Rebuttal**
> >
> > Thanks for the detailed rebuttal. The added ablation generally addressed the corresponding concern on convincingness of experimentation. The concerns on the first two points still remains in my view, so I would keep my current rating due to the questionable rationales but I encourage the authors to further improve their work since the general idea is interesting and novel.

---

> ### Author Response · Authors · 2025-08-06
> **A gentle reminder for author-reviewer discussions**
>
> Dear Reviewer bFhU,
>
> Thanks for your thorough reviews and helpful suggestions. Your instructions really inspired us and help to make this work more solid. We have tried our best to address the issues you mentioned in the review. We will definitely follow your suggestions and integrate the new results in the next version of our paper. Could you please let us know if there are any remaining concerns, or anything we can do to further improve our work?
>
> Warm regards,
>
> The authors.

---

> ### Author Response · Authors · 2025-08-06
> **We respect your comments but could you please elaborate a little bit?**
>
> Dear Reviewer bFhU,
>
> We are so sorry that our rebuttal does not fully address your concerns. Could you please slightly elaborate what are the questionable rationales? On the existence of the mental reward or the inappropriate modeling of it? Or perhaps the lack of evidence to support it? It would be very helpful for us to respond to your concerns.
>
> Meanwhile, we totally respect your decision because the "mental reward" is indeed a new concept, and it might require a progress to fully accepting it. Anyway, we thank for your constructive suggestions that help to improve our work. We hope that you can give us a chance to explain for your remaining concerns. Thank you!
>
> Best wishes,
>
> The authors.

---

> > ### Comment · Reviewer_bFhU · 2025-08-07
> > **Reviewer bFhU's Second Comment on Rebuttal**
> >
> > For the mental reward definition, it is a rather vague definition which partially takes the essence of the definition of user preferences (not only topic or interest level preference, but also presentation style and other behavioral and contextual preferences) and user satisfaction that closely related to the engagement metrics. On the other hand, if it is well-defined, then it would be much more convincing if it discovers a set of mental reward system with explicit and general semantics that changes the overall evaluation metrics of recommender systems.

---

> > > ### Author Response · Authors · 2025-08-07
> > > **Explanations on why the definition of "mental reward" should be vague.**
> > >
> > > Dear Reviewer bFhU,
> > >
> > > Thanks for your precious time and giving us the opportunity to explain for your concerns. For brevity, we only want to make the following two points.
> > >
> > > ### **Point 1: Existing user preference modelings are not adequate, the gap with users' real preferences still exists.**
> > >
> > > As you have mentioned, there are many existing approaches to model user preference from different aspect (e.g., topic or interest level, contextual level). However, such explicit modelings often focus on specific aspects. and combining them together to approximate user's real preference is almost impossible given the complexity of recommender systems. As a result, there is still a significant gap between users' feedback modeling and their real preference. **Note that user preference itself is a general concept without a clear definition.**
> > >
> > > ### **Point 2: To close the gap, the definition of "mental reward" needs to be vague.**
> > >
> > > Since the explicitly defined user preference modeling methods have nature limitations, we explore an alternative way to approximate users' real preferences. As we state in the paper, the mental reward is defined as "a summary of user's feelings in each recommendation step", which allows us to **learn** it indirectly from users' behaviors. If there exists an explicit definition of the mental reward,  we should have a way to **calculate** it based on its definition. Then, we can always find counter examples to prove the existence of the gap with user's preference, because the concept of user's real preference itself is vague.
> > >
> > > To help with understanding the concept of "mental reward", we can refer to how the LLMs are aligned with humans (including their political values, social values, life values, etc.). **We can't clearly define what are the LLMs aligned with, right?** The reward model used in RLHF is also a summary of human values, which inspired us to develop the mental reward.
> > >
> > > However, we agree with the reviewer that there might be a set of mental reward systems in user's mind, and dicovering such systems would help to build more comprehensive evaluation metrics of recommender systems. We will continue to explore this direction as the reviewer suggested. Considering that MTRec is the first work trying to close the gap using a reward model based approach, we believe that our work makes a step towards this goal.
> > >
> > > ### **Action: We will summarize the discussions with Reviewer bFhU and try to describe the concept of mental reward more clearly in the next version of our paper.**
> > >
> > > Thanks again Reviewer bFhU, we hope that the above explanations can be taken into consideration when you make the final evaluation of our work. Meanwhile, we would love to here from you again and be prepared to respond to your further comments.
> > >
> > > Best wishes,
> > >
> > > The authors.

---

> > > > ### Comment · Reviewer_bFhU · 2025-08-08
> > > > **Third Comment on Rebuttal**
> > > >
> > > > Thanks for the explanation, I understand the difficulty of achieving a well-defined mental reward and appreciated the effort on the extended ablation in the rebuttal which partially addressed my concern. Though the concerns in the discussion are only partially addressed in my view and there is still room to improve the convincingness of the paper. I believe the current revised state of the work is reaching the bar of NeurIPS so I am raising the rating to 4.

---

> > > > > ### Author Response · Authors · 2025-08-08
> > > > > **Thank you Reviewer bFhU**
> > > > >
> > > > > Dear Reviewer bFhU,
> > > > >
> > > > > Thank you so much for your acknowledgement on our contribution. We will follow your insightful suggestions to improve our work.
> > > > >
> > > > > Best wishes,
> > > > >
> > > > > The authors.

---

### Official Review · Reviewer_7bFs · 2025-07-03

**Clarity:** 3
**Significance:** 3
**Originality:** 4
**Rating:** 5
**Confidence:** 4

**Summary:**

This paper proposes a novel framework designed to align recommendation models with users' intrinsic preferences. The key innovation involves the construction of a mental reward model that quantitatively captures user satisfaction levels. To effectively learn this model, it introduces a distributional inverse reinforcement learning methodology. The optimized mental reward model subsequently serves as a guiding mechanism for training the recommendation system. Extensive experiments are conducted to verify the effectiveness of the proposed method.

**Questions:**

1. Regarding Appendix A.3 setting the number of quantiles N=10, it does not explain why 10 quantiles are chosen instead of 5 or 20 (e.g., the balance of probability interval coverage). Additionally, the logic behind the non-uniform distribution of quantile levels (e.g., 0.2, 0.3, ..., 0.9) is not clarified (e.g., the theoretical basis for dense sampling in medium-high probability intervals). It is recommended to supplement comparative experiments with N=5 or 20 to demonstrate robustness.

2. Concerning the engineering details of industrial deployment, the online experiment mentions integrating the alignment loss with the DCN model but does not specify the real-time calculation latency of mental rewards (e.g., whether it affects the recommendation response time).

3.  The evaluation section indirectly validates the model through counterfactual mental rewards, but lacks a correlation analysis (e.g., Pearson coefficient analysis) between reward values and explicit user metrics (such as viewing duration and revisit rate). This may weaken the core assumption that "rewards reflect real preferences."

**Ethical Concerns:**

["NO or VERY MINOR ethics concerns only"]

**Limitations:**

Yes

**Quality:**

3

**Strengths And Weaknesses:**

Strengths
1. The Idea of this paper is interesting and novel. Departing from conventional recommendation paradigms that treat implicit feedback (e.g., clicks) as deterministic preferences, this work proposes modeling user behavior as "a stochastic process of mental reward maximization".
2. To address the limitation of conventional inverse reinforcement learning (IRL) methods in capturing reward distributions, this work introduces the Distributional Inverse Reinforcement Learning algorithm (QR-IQL).
3. Comprehensive experiments and analysis are provided, including offline and online experiments, as well as testing on a large-scale industrial recommendation platform.

Weakness:
it still lacks a systematic method to thoroughly evaluate the learned mental reward model.

---

> ### Author Rebuttal · Authors · 2025-07-30
>
> Thanks for your appreciation and helpful suggestions to our work. Please see our responses below.
>
> **Q1: Why the number of quantiles N=10 is chosen instead of 5 or 20?**
>
> **A1:** Thanks for raising this question. Intuitively, a larger N (more quantiles) provides more flexibility to model the underlying distribution of $Q_\lambda$, however, it may also bring overfitting issues. We agree that searching for the optimal N is necessary. Therefore, we add a new set of experiments on the Electronic dataset over three models, as shown below.
>
> |    Model    | AUC (N=5)| AUC (N=10)| AUC (N=15)| AUC (N=20)| NCIS (N=5)| NCIS (N=10)| NCIS (N=15)| NCIS (N=20)|
> | :----------- | :-----------: | :-----------: | :-----------: | :-----------: | :-----------: | :-----------: | :-----------: | :-----------: |
> | DeepFM-MTRec      | 0.8462      |**0.8468**    | 0.8467       | 0.8463      | 0.8960      |0.8961      | 0.8960       |**0.8975**      |
> |DIN-MTRec               | 0.8544      |0.8542     | 0.8544      | **0.8547**      | 0.8722      | 0.8728     | **0.8733**       |0.8731      |
> | LinRec-MTRec         | 0.8593      |0.8594     | **0.8597**      | 0.8589      | 0.9773      |0.9782      | 0.9780       |**0.9785**      |
>
> From the above table, we can see that different choices of N do not result in significant changes in terms of AUC and NCIS.  In addition, N=5 performs the worst, perhaps due to limited flexibility to represent the underlying quantile distribution.
>
> We also test different choices of N on Virtual Taobal over two RL-based methods. The results are reported below.
>
> |    Model    | Averaged CTR (N=5)| Averaged CTR (N=10)| Averaged CTR (N=15)| Averaged CTR (N=20)|
> | :----------- | :-----------: | :-----------: | :-----------: | :-----------: |
> | PPO-MTRec      | 0.665      |0.678   | 0.686       | **0.688**      |
> |SAC-MTRec               | 0.885      |0.909     | 0.915      | **0.917**      |
>
> Surprisingly, we obverve a clear trend of improving performance, with respect to N, of both PPO and SAC. These results show that a larger number of quantiles helps to learn more accurate mental reward models. Note that this trend is not that obvious in supervised learning models (refer to the first table). We hypothesis that it is because RL-based methods are more sensitive to rewards.
>
>
> **Q2: Whether the calculation of mental rewards affects the recommendation response time?**
>
> **A2:** In fact, according to Algorithm 2 in Appendix A.3, we only calculate the mental rewards during training. There is no extra computation that might increase the response time during testing. We are sorry for the misunderstanding caused and will state more clearly in the paper.
>
> **Q3: Whether the calculation of mental rewards affects the recommendation response time?**
>
> **A3:** Thanks for your insightful suggestion! Since there is no direct way to evaluate the mental reward model, computing correlations is an effective approach. In the first figure of Figure 3, we report the correlation between users' mental rewards and the lengths of their interaction with the recommender system. We can see that users' mental rewards gradually decrease with the number of interaction steps increases. This aligns with the intuition that people might lose interest and get tired after consuming enough recommendation contents.
>
> Thanks again for reviewing our paper, we hope that the above rebuttal can address your concerns.

---

> ### Author Response · Authors · 2025-08-06
> **A gentle reminder for author-reviewer discussions**
>
> Dear Reviewer 7bFs,
>
> Thank you for your appreciation and valuable support to our work. Your thorough review and insightful suggestions really make our work better. As the deadline for the rebuttal phase approaching, could you please let us know if there are any remaining concerns or further suggestions?
>
> Warm regards,
>
> The authors.

---

### Official Review · Reviewer_pp7H · 2025-07-05

**Clarity:** 2
**Significance:** 2
**Originality:** 2
**Rating:** 3
**Confidence:** 4

**Summary:**

This paper proposes MTRec, a sequential recommendation framework that introduces a mental reward model to better align recommendations with users’ real preferences. The mental reward is defined as a latent signal reflecting internal user satisfaction, and is learned from behavioral data via a distributional inverse reinforcement learning method called Quantile Regression Inverse Q-learning (QR-IQL). The learned reward is then used to provide auxiliary supervision for training recommendation models. Experiments are conducted on public datasets, a simulated environment (Virtual Taobao), and an industrial short-video platform, showing improvements in offline metrics and online engagement.

**Questions:**

Q1. The paper assumes that users are maximizing their cumulative “mental rewards,” but this concept is treated as a latent variable without clear grounding. How sensitive is the proposed method to this assumption, and is there any empirical evidence or user-behavior analysis to support it?

Q2. How does MTRec conceptually and practically differ from existing reinforcement learning-based recommendation frameworks? Could the authors clarify the complementary or novel aspects compared to those methods?

Q3. The mental reward model is used as an auxiliary supervision signal, but its interpretability remains unclear. Have the authors considered any methods to analyze or visualize what the model has learned?

**Ethical Concerns:**

["NO or VERY MINOR ethics concerns only"]

**Limitations:**

yes

**Paper Formatting Concerns:**

The reference formatting requires revision to meet academic standards.

**Quality:**

3

**Strengths And Weaknesses:**

Strengths

S1. The paper addresses a core challenge in recommendation — the discrepancy between implicit feedback and true user preferences — and proposes a novel approach that attempts to model latent user satisfaction signals directly.

S2. The proposed QR-IQL method offers a technically interesting contribution by capturing the stochastic nature of user feedback through a distributional IRL formulation, extending prior work on inverse Q-learning.

Weaknesses

W1. The concept of mental reward is central but underdefined. The paper lacks theoretical grounding or empirical validation to support that the learned reward meaningfully reflects user satisfaction.

W2. The learned reward model lacks interpretability and analysis. There is limited insight into what the model captures, how it varies across users or items, or whether it corresponds to known signals of user preference.

W3. The experimental comparison omits several recent and relevant RL-based recommendation models that similarly aim to optimize long-term user satisfaction or retention, such as Decision Transformer-based frameworks or offline RL approaches. Without comparing to these baselines (e.g., [1-4]), it is difficult to assess whether the proposed method offers a significant advantage over the current state of the art.

Ref:

[1] Sequential recommendation for optimizing both immediate feedback and long-term retention, SIGIR 2024.

[2] User retention-oriented recommendation with decision transformer, WWW 2023.

[3] Maximum-entropy regularized decision transformer with reward relabelling for dynamic recommendation, KDD 2024.

[4] Causal decision transformer for recommender systems via offline reinforcement learning, SIGIR 2023.

---

> ### Author Rebuttal · Authors · 2025-07-30
>
> Thanks for your helpful suggestions to our work. Please see our responses below.
>
> **Q1: On the definition of the mental reward.**
>
> **A1:** Thanks for raising this question. In the third paragraph of the Introduction, we define the user's mental reward as "the summary of her private feelings about how she is satisfied". We also use multiple examples to illustrate this concept. However, we agree with you that we did not provide a very specific definition of the mental reward, because it is essentially **a highly abstract concept and can be interpreted in different ways depending on the scenario contexts**. For example, a user might click on a news article because of its attractive headline, but end up feeling uncomfortable after reading the content. Here, the mental reward is a reflection and quantification of how the user feels uncomfortable.
>
> On the other hand, despite its definition, the mental reward **does exist**, and it lies between user's implicit feedback (e.g., click) and her real preferences. Our work aims to narrow this gap by using the mental reward as an effective tool (and the experimental results demonstrate this). In summary, our work makes an important step towards understanding the gap between users' implicit feedback and their real preference. We contribute a novel and theoretically solid algorithm to uncover the mental rewards and demonstrate its effectiveness by extensive experimental results.
>
> **Q2: How sensitive is the proposed method to the assumption that users are maximizing their cumulative mental rewards?**
>
> **A2:** This is an insightful question. Actually, this assumption is the basic motivation of using inverse RL, where we aim to seek a reward function that rationalize users' behaviors. If the users are not maximizing their mental rewards, which means that learned mental rewards are **noises**, they should not bring improvements to the recommendation models. However, our experiments (including offline, online, real-world deployment) demonstrate the effectiveness of the learned mental rewards, which in turn **justifies the correctness of this assumption**.
>
> **Q3: How does MTRec conceptually and practically differ from existing reinforcement learning-based recommendation frameworks?**
>
> **A3:** **MTRec is actually a complementary to RL-based methods.** RL-based methods heavily rely on accurate rewards to learn good policies, yet existing approaches simply use user implicit feedback (e.g., click) as rewards. Due to the gap between the implicit feedback and their true preferences, the implicit feedback may not be the optimal reward signal. MTRec makes a non-trivial correction to the reward signals, which significantly improves the performance of RL-based methods. Please refer to Section 5.2 for more details.
>
> **Q4: Have the authors considered any methods to analyze or visualize what the model has learned?**
>
> **A4:** If we understand corretly, this question is related to Q1. Perhaps answer A1 could clarify some of the concerns. In a word, the mental reward model learns to summarize and differentiate user's feelings after she takes an action. For example, if a user clicks and reads a piece of news, how much does she like it actually? In Section 5.3, we visualize some analysis on the mental reward model. We can see that the mental rewards actually correlated with users' behaviors, which also aligns with our intuitions. Although we don't know exactly how the learned mental reward correspond to users' feelings (e.g., like, neutral, dislike, angry, favorite), it indeed provides useful information to the recommender systems and helps to better understand the users.
>
> **Q5: Comparisons to Decision Transformer based methods.**
>
> **A5:** Thanks for raising this question. In the paper we consider online RL setting and test MTRec on PPO and SAC. As the reviewer mentioned, it is also interesting to test MTRec on offline RL methods such as Decision Transformers. We use the dataset containing 100,000 user-RS interaction trajectories, as described in Section 5.2, to train a DT model (including 2 Transformer layers, 8 heads, and an embedding size of 128) for 5 epochs, and test it the same way with PPO and SAC in the online Virtual Taobao environment. The comparative results are shown below.
>
>
> | Model      | Averaged CTR |
> | :---        |    :----:   |
> |   PPO  | 0.5435       |
> | PPO-MTRec   | 0.6782       |
> |   SAC  | 0.7055       |
> | SAC-MTRec   | 0.909       |
> |   DT  | 0.4816      |
> | DT-MTRec   | 0.5472       |
>
> It can be observed that the original DT performs worst across all models, since DT in trained in offline setting while PPO and SAC are trained in online setting. Therefore, it is not a really fair comparison. However, we can see that DT-MTRec still improves DT, which demonstrates that MTRec indeed brings more useful information than simply treating click as reward signal.
>
> Note that all of the four references mentioned by the reviewer are based on DT, therefore we only test the original DT and combine it with MTRec. In the next version of our paper, we will discuss more about the differences between our work and existing RL-based methods, including but not limit to the references mentioned by the reivewer.
>
> Thanks again for your time spent on our paper. We would be very grateful if you could re-evaluate our work based on our rebuttal.

---

> > ### Comment · Reviewer_pp7H · 2025-08-03
> >
> > Thank you for the response. I find the attempt to model mental reward a very interesting endeavor, and indeed, we can cite many examples to indirectly support its existence. However, this is also where my concerns arise: while the examples are compelling, it remains uncertain whether the underlying assumption holds beyond these specific cases. If the assumption fails in other scenarios, such modeling could lead to potential risks or losses. These are some of my reservations about the study, reflecting my concerns about the practical applicability of the proposed solution. That said, I also encourage the authors to continue exploring this promising direction.

---

> > > ### Author Response · Authors · 2025-08-03
> > > **Perhaps the "No Free Lunch" theory explains for us**
> > >
> > > Dear Reviewer pp7H,
> > >
> > > Thank you for your feedback on our rebuttal. We really appreciate your insightful comments and encouragement to us.
> > >
> > > Regarding to your remaining concerns, we totally agree with you that our assumption may not work in some scenarios. But please let us argue for this. In fact, according to our experinences, there is no single model that works for most recommendation scenarios, which corresponds to the "**No Free Lunch**" theory.
> > >
> > > The reason behind this is the diversity and complexity of the recommendation scenarios. Yet, we tried our best to make MTRec cover as many cases as possible. For instance, we propose a general modeling approach that covers a wide range of scenarios in sequential recommendation. We have demonstrated that our work brings significant advantages to both classification-based models and RL-based models. We also test MTRec on a large-scale industrial platform.
> > > **These evidences demonstrate the practical impact of MTRec.**
> > >
> > > However, we value the reviewer's concerns and will continue to explore this direction. Considering that MTRec is the first work to study the mental reward, we really wish that the reviewer could take our responses into consideration when making the final decisions on our work. Thanks again!
> > >
> > > Best wishes,
> > >
> > > The authors

---

> > > ### Author Response · Authors · 2025-08-09
> > > **The last gentle reminder for author-reviewer discussion**
> > >
> > > Dear Reviewer pp7H,
> > >
> > > As the deadline comes in a few of hours, we want grasp this last chance to discuss with you and try to win your support.
> > >
> > > We will be ready to respond until the deadline. So, please do let us know if there are remaining concerns and we will respond ASAP.
> > >
> > > Anyway, we sincerely thank you for your constructive suggestions. We will take those and continue to improve our paper.
> > >
> > > Best wishes,
> > >
> > > The authors.

---

> ### Author Response · Authors · 2025-08-08
> **Summary of the discussion and follow up actions**
>
> Dear Reviewer pp7H,
>
> Thanks again for your constructive suggestions and precious time spent on our paper. As the deadline of the author reviewer approaching, we want to briefly summarize the follow up actions and try to win your support on our work.
>
> ### **Action 1**:
> Based on your comments on the mental reward, we will state more clearly on the definition of the mental reward and explain how this new concept relates to existing concept such as user preference modeling, user satisfaction, explicit and implicit preference. However, we may still argue that the definition of the mental reward needs some flexibility. Because we want to use it to approximiate *user's real preference*, which itself is a vague concept.
>
> ### **Action 2**:
> We will discuss about the sensitivity of our assumptions and continue to explore in this direction. For example, if there are only part of the users are rational, how should our model respond to improve robustness. We thank you for raising this interesting point.
>
> ### **Action 3**:
> We will discuss the related works you mentioned in the reviews and integrate new comparative experimental results in the Experiment section. This would definitely strengthen our claims. Thanks for reminding us of these works.
>
> ### **Action 4**:
> As we have mentioned in the paper, our model brings more improvements on user's long-term engagement (measured by NCIS), instead of AUC. However, we will follow the reviewer's suggestions and state more clearly on the potential risk of losses. Thanks for raining this important point.
>
> We hope that our responses help to clarify the misunderstandings of the original paper. We would be so grateful if you consider to re-evaluate our paper based on the responses and follow up actions.
>
> Thank you sincerely,
>
> The authors.

---

### Author Response · Authors · 2025-08-01
**Summary of reviews and general responses**

Dear Reviewers and ACs,

We sincerely thank you for your evaluations and suggestions to our work. That's really helpful! We want to give a summary of reviews and responses so that to facilitate the author-reviewer discussion.

### **Pros**:

1. Our work targets a novel and fundamental research problem in sequential recommendation (**All reviewers**).

2. Our core algorithmic contribution-the QR-IQL algorithm-is technical solid and clearly presented (**All reviewers**).

3. Comprehensive experimental results demonstrate the advantage of MTRec (**Reviewer 7bFs and nSej**).

4. Real-world experiments on industrial recommendation platforms highlight the practical impact of our work (**Reviewer 7bFs, bFhU and nSej**).

We are very grateful for reviewers appreciation on our work.

### **Cons**:

**Q1: The interpretation of the mental reward.**

**A1**: We agree that the concept of the mental reward might be hard to capture, because it is **completely new and abstracted concept, and perhaps related to other sciences such as psychology (as is mentioned by Reviewer nSej)**. We illustrated this concept by multiple domain examples and discussions. We also included an experimental section to show the correlations between the mental rewards and users' actual behaviors. Please refer to our responses to Reviewer pp7H and bFhU. We hope that our efforts could help to strengthen the understanding of this new concept.

**Q2: Supplementary experiments.**

**A2**:  Reviewer pp7H suggests to compare our method with Decision Transformer-based baselines, while Reviewer bFhU suggests to add more ablation studies. We agree with the reviewers that these new experiments would strengthen the claims of our work. Therefore, we provide the new experimental results and analysis in the rebuttal (please refer to our responses to Reviewer pp7H and bFhU). We will discuss the works suggested by reviewers and integrate new experimental results in the next version of our paper.

At last, we thank all the reviewers and ACs again for your precious time spent on our paper. We hope that our rebuttal could address your cocnerns. We look forward to futher discussions with reviewers in these days. So, please let us know if there are any remaining concerns.

---

### Note · Authors · 2025-08-11

Dear Reviewers, AC and SAC,

Thanks for your precious time! To facilitate your discussion, we briefly summarize the resolved and remaining issues for your reference.

### **Resolved:**
1. Comparisions with DT-based methods (**Reviewer pp7H**).

2. Ablations on the number of quantiles (N), the weights of alignment losses and computational cost (**Reviewer 7bFs and bFhU**).

3. Discuss the relation between mental reward to other sciences (**Reviewer nSej**).

4. The interpretation of the proposed "mental reward" concept. **Reviewer pp7H and bFhU** raised this issue in their initial reviews. After the rebuttal, as I quote, Reviewer pp7H *finds the attempt to model mental reward a very interesting endeavor*, Reviewer bFhU understands *the difficulty of achieving a well-defined mental reward* and raised her rating, authough she mentioned that there is still room to improve the definition of the mental reward (we agree on this).

### **Remaining:**
1. If the assumption on the mental reward fails in other scenarios, such modeling could lead to potential risks or losses (**Reviewer pp7H**).

>We explained this using the "No Free Lunch" theory in the rebuttal to Reviewer pp7H. We agree that MTRec may not work for all recommendation scenarios, but as we have demonstrated in the paper, it at least works for many popular classification-based models and RL-based models. We also demonstrated its effectiveness in online platform. **We believe that these evidences strongly support the correctness of the assumption**. However, as the reviewer suggested, we will continue to explore in this direction and consider more realistic scenarios.

2. Calculation latency of mental rewards and correlation analysis (**Reviewer 7bFs**).

>Please refer to our rebuttal to Reviewer 7bFs (Sorry for the wrong title of Q3. It should be related to "correlation analysis".).

### **Conclusion**:

All reviewers identify our work as **interesting, novel** and **technically solid**. The reviewers are generally satisfied with the supplementary experimental results. We believe that our work opens a new direction and makes an important step towards understanding/closing the gap between the users' feedback and their real preferences.  We will take all the reivewer's suggestions and improve our paper, as we have committed in the rebuttal.

We sincerely thank all reviewers and ACs. No matter what the outcome, discussions with you indeed make this work better. Thank you!

Best wishes,

The authors.

---

### Decision · Program_Chairs · 2025-09-17

**Decision:**

Accept (poster)

**Comment:**

This paper proposes MTRec, a sequential recommendation framework that introduces the concept of a “mental reward” to bridge the gap between implicit feedback and true user preferences (explicit feedback). The method leverages a distributional inverse reinforcement learning approach (QR-IQL) to learn mental reward models and demonstrates their utility in recommender systems. Reviewers acknowledged the novelty of the problem formulation, the technical soundness of the algorithm, and the comprehensive evaluations across public datasets, simulated environments, and an industrial platform. The primary concerns centered on the vague definition of “mental reward,” limited interpretability, and the need for stronger comparisons and ablation analyses. The rebuttal and discussion phases helped alleviate these issues, as the authors clarified the interpretation of mental reward, added ablations (quantiles, loss weights, and computational cost), and extended comparisons with Decision Transformer baselines. Given the paper’s strong empirical results, clear algorithmic innovations, and demonstrated practical impact, I recommend acceptance.